# Control Efficacy of Entomopathogenic Fungus *Purpureocillium lilacinum* against Chili Thrips (*Scirtothrips dorsalis*) on Chili Plant

**DOI:** 10.3390/insects13080684

**Published:** 2022-07-28

**Authors:** Cheerapha Panyasiri, Sumalee Supothina, Sukitaya Veeranondha, Rungtiwa Chanthaket, Tanapong Boonruangprapa, Vanicha Vichai

**Affiliations:** National Center for Genetic Engineering and Biotechnology, Thailand Science Park, Pathum Thani 12120, Thailand; sumaleeu@biotec.or.th (S.S.); sukitaya@biotec.or.th (S.V.); rungtiwa.cha@biotec.or.th (R.C.); tanapong.boo@biotec.or.th (T.B.); vanicha@biotec.or.th (V.V.)

**Keywords:** *Purpureocillium lilacinum*, *Beauveria bassiana*, *Scirtothrips dorsalis*, biocontrol

## Abstract

**Simple Summary:**

Chili thrips (*Scirtothrips dorsalis*) is an important pest of chili crops and a major vector of viral plant pathogens. Due to the widespread outbreak of thrips, chemical insecticides have been heavily used in the last few decades. To reduce the utilization of chemical pesticides, alternative biocontrol agents such as entomopathogenic fungi have been screened against the thrips. Laboratory screening revealed that 2 insect fungi isolates, *Purpureocillium lilacinum* TBRC 10638 and *Beauveria bassiana* BCC48145 were the most effective isolates against chili thrips. The fungus, *P. lilacinum* TBRC 10638 exhibited the highest efficacy against chili thrips in greenhouse and field trials and thus would be developed as a thrips control agent.

**Abstract:**

In a laboratory assay, it was shown that *B. bassiana* BCC48145, BCC2660, and *P. lilacinum* TBRC10638 were the three strains that exhibited the highest insecticidal activity against chili thrips, causing 92.5% and 91.86% and 92.3% corrected mortality, respectively. The fungi *B. bassiana* BCC48145 and *P. lilacinum* TBRC10638 were selected for greenhouse spraying. Cytotoxicity test of the extracts from both fungi evaluated against 4 animal cell lines: KB; human oral cavity carcinoma, MCF7; human breast adenocarcinoma, NCI-H187; human small cell lung carcinoma and GFP-expressing Vero cells, showed none-cytotoxic to all cell lines. An efficacy validation in the greenhouse showed that *P. lilacinum* TBRC 10638 was more effective than *B. bassiana* BCC48145 and could control the thrips up to 80% when using the fungus at 10^8^ spores/mL. The LC_50_ values of *P. lilacinum* TBRC 10638 against chili thrips based on total thrips count from two experiments were 1.42 × 10^8^ and 1.12 × 10^7^ spores/mL when the fungal spores were sprayed once a week. The optimal concentration of *P. lilacinum* TBRC 10638 spores for effective control of chili thrips was determined at 1.41 × 10^9^ spores/mL. The average efficacy of *P. lilacinum* TBRC 10638 for thrips control from 3 field trials was 30.08%, 14.39%, and 29.92%. This result was not significantly different from that of the chemical insecticide treatment group, which showed efficacy at 19.27%, 14.92%, and 19.97%. Furthermore, there was no difference in productivity among the different treatment groups. Our results demonstrated that *P. lilacinum* TBRC 10638 is a promising biocontrol agent that could be used as an alternative to chemical insecticide for controlling chili thrips.

## 1. Introduction

Chili thrips (*Scirtothrips dorsalis*) is an important pest that affects more than 100 plant taxa around the world; including vegetable crops such as cucumber, pepper, and eggplant; fruit crops such as grape, lemon, and mangosteen; and ornamental crops such as rose and green buttonwood [1,2]. Chili thrips feed on all parts of plants, especially young leaves, buds, and fruits [3], on which the insect inflicts damage by extracting the contents of epidermal cells leading to necrosis of tissue. Thrips feeding on the leaves cause changes in tissue color from silver to brown and black. Chili thrips also transmit 7 viral pathogens including chili leaf curl (CLC) virus, peanut necrosis virus (PBNV) [4,5], and tobacco streak virus (TSV) in groundnut crops, as reported in India [6]. Recently, in Thailand chili thrips were reported as a vector of three tospoviruses i.e., melon yellow spot virus (MYSV), watermelon silver mottle virus (WsMoV), and capsicum chlorosis virus (CaCV) in field crops [7].

Many countries avoid insect pests by employing greenhouse cultivation, which nevertheless cannot prevent infestation by tiny insects such as whiteflies, mealybugs, and thrips. Normally, thrips are controlled by contact insecticides, but their habitats on plants especially on flowers or buds are well protected from chemical spray. Since it is difficult to achieve effective control of these insects, farmers are obligated to use high dosages of insecticides and increase spraying frequency [8]. As a consequence, insecticidal resistance has been reported in many countries. Thrips resistance to spinosad, which is a commonly used insecticide, has been reported in the USA [9], Australia [10], and China [11]. In Iran, it has been reported that *Thrips tabaci* populations are resistant to several classes of insecticides including diazinon and dichlorvos (organophosphorus), permethrin, and cypermethrin (pyrethroids), acetamiprid (neonicotinoids), spinosad (spinosyns) and azadirachtin (triterpenoids) [12].

Currently, Integrated Pest Management (IPM) systems for greenhouses utilize chemicals, parasites, predators, and entomopathogenic agents such as bacteria, viruses, and fungi. The entomopathogenic fungi are increasingly used for pest control in greenhouses, which provide a suitable environment for fungal growth. The entomopathogenic fungi that are widely used as biological controls include *Beauveria bassiana* against a broad range of insect pests such as diamondback moth (*Plutella xylostella*) [13], green peach aphid (*Myzus persicae*), whitefly (*Bemisia tabaci*), cicada (*Lacobiasca formosana*) [14] and cigarette beetle (*Lasioderma serricorne*) [15] *Metarhizium anisopliae* against sweet potato weevil (*Cylas formicarius*) [16], cotton bollworm (*Helicoverpa armigera*) [17] and termite (*Odontotermes obesus*) [18]; *Paecilomyces fumosoroseus* against whitefly (*Bemisia argentifolii*) [19] and tomato thrips (*Ceratothripoides claratris*) [20]; and *Verticillium lecanii* usually against aphid (*Aphis gossypii*) and whitefly (*Trileurodes vaporariorum*) [21]. For thrips control, there have been reports on the pesticidal activity of *P. fumosorosea* against *C. claratris* [20], *M. anisopliae* against *T. tabaci* [22], and *B. bassiana* and *Purpureocillium lilacinus* against *Thrips palmi* [23,24].

In this study, entomopathogenic fungi selected from the previous works were evaluated for the control of chili thrips (*S. dorsalis*) in laboratory, greenhouse, and field trials. The median lethal concentration (LC_50_) against the insect and cytotoxicity against mammalian cells were determined to assess the potential of these fungi for application in the field.

## 2. Materials and Methods

### 2.1. Entomopathogenic Fungal Isolates

Eight entomopathogenic fungal isolates selected from previous screening against tomato thrips (*C. claratris*) [20] and aphid (*M. pericae*) [25] (Table 1) were obtained from BIOTEC Culture Collection (BCC). They proved to be highly effective against both homopteran insects. The selected fungal isolates were cultured on Potato Dextrose Agar (Difco^TM^, Sparks, MD, USA) at 25 °C for 7 days. Fungal spores were harvested from each fungal colony by adding 5 mL of 0.1% Tween 80 (Sigma-Aldrich, St. Louis, MO, USA) to the colony and scraping with a spatula. The spore suspension was transferred to a 15 mL centrifuge tube, mixed for 1 min, and then filtered using sterile gauze to separate hyphae from spore suspension. Spore concentration was adjusted to 2 × 10^8^ spores/mL for bioassay against thrips.

### 2.2. Fungal Cultivation and Extraction

*P. lilacinum* TBRC10638 and *Beauveria bassiana* BCC48145 were maintained on potato dextrose agar at 25 °C for 14 days. Each agar culture was cut into 2 × 2 cm^2^ with a sterilized surgical knife and inoculated in a 250 mL Erlenmeyer flask containing 30 g rice grain added with 12 mL tap water. The solid cultures of both fungi were incubated at 2 °C for 14 days. Then, conidiospores of each strain were re-suspended in 25 mL of 0.1% Tween 80 (10^9^ spores/mL) and these suspensions were diluted to the concentrations of 2 × 10^7^–2 × 10^8^ spores/mL as inoculum and starter culture.

The production of conidiospores and mycelia culture were conducted by using the same starter culture. For conidiospores preparation, 12 mL of starter culture was transferred into 10 plastic bags containing 200 g of rice grains autoclaved with 80 mL tap water and incubated at 25 °C for 10 days. Conidial spores were harvested and washed with 0.1% Tween 80 at least twice. Then, the wet spores were soaked in methanol for 2 days. Subsequently, the methanol extract of spores was extracted with ethyl acetate. Crude extracts were used for cytotoxicity test as described by Luangsa-ard et al., (2009) [26].

The mycelial cultivation of each fungal strain was carried out in potato dextrose broth (PDB). Twenty-five milliliters of starter culture were inoculated into a 1000 mL Erlenmeyer flask containing 250 mL PDB and incubated on a rotary shaker (200 rpm) at 25 °C for 7 days. Afterward, fungal culture was filtrated to separate wet mycelia and culture broth, which were then extracted for cytotoxicity test [26].

### 2.3. Cytotoxicity Test

The toxicity of *P. lilacinum* TBRC10638 and *B. bassiana* BCC48145 extracts was evaluated by resazurin microtiter plate assay [27] against animal cell lines: KB human oral cavity carcinoma; MCF7, human breast adenocarcinoma; NCI-H187, human small cell lung carcinoma; and GFP-expressing Vero cells, African green monkey kidney cell line transfected with plasmid carried *gfp* gene (pEGFP-N1, Clontech). KB (ATCC CCL-17), MCF7 (ATCC HTB-22), NCI-H187 (ATCC CRL-5804), and Vero (ATCC CCL-81) cell lines were obtained from the American Type Culture Collection (ATCC). KB and MCF7 cell lines were grown and maintained in Minimal Essential Medium (MEM) supplemented with 10% heat-inactivated fetal bovine serum, 1 mM sodium pyruvate, 0.1 mg/mL insulin, 0.1 mM non-essential amino acid, and 1.5 g/L sodium bicarbonate. NCI-H187 cells were grown and maintained in RPMI-1640 supplemented with 15% heat-inactivated fetal bovine serum, 1 mM sodium pyruvate, 2.5 g/L Glucose, and 2.2 g/L sodium bicarbonate. GFP-Vero cells were grown and maintained in MEM supplemented with 10% heat-inactivated fetal bovine serum, 1 mM sodium pyruvate, 0.1 mM non-essential amino acid, 2.2 g/L sodium bicarbonate, and 0.8 mg/mL geneticin. All lines were incubated at 37 °C in a humidified incubator with 5% CO_2_.

Cytotoxicity assay was performed in 384-well plates in triplicate. Each well was added with 5 µL of fungal extract at 500 µg/mL in 5% dimethylsulfoxide (DMSO) and followed with 45 µ of KB, MCF7, NCI-H187, or GFP-Vero cell suspension at 2.2 × 10^4^, 3.3 × 10^4^, 6.7 × 10^4^ or 3.3 × 10^4^ cells/mL, respectively. In place of fungal extract, anti-cancer drugs were used as positive controls: ellipticine for GFP-Vero, ellipticine, and doxorubicin for KB and NCI-H187 and tamoxifen and doxorubicin for MCF7, whereas 5% DMSO was used as a negative control.

For GFP-Vero, cells were incubated at 37 °C in a humidified incubator with 5% CO_2_ for 4 days. Fluorescent signals were measured on day 0 and day 4 using the excitation and emission wavelengths of 485 and 535 nm., then the signal of day-4 was subtracted from that of day-0 before calculation. In the assays using other cell lines, cells were incubated at 37 °C in a humidified incubator with 5% CO_2_ for 3 days (for KB and MCF7) and 5 days (for NCI-H187). Afterward, 12.5 µL of 62.5 µg/mL resazurin solution was added to each well and the plates were further incubated at 37 °C for 4 h. Fluorescence was measured at 530 nm excitation and 590 nm emission wavelengths. The signals were then subtracted with a blank before calculation. The % cytotoxicity was calculated from subtracted fluorescent signals from extract-treated wells compared to the negative control. The % inhibition at 50% was used as a cut-off for cytotoxicity.

### 2.4. Insect Rearing

Chili thrips (*S. dorsalis*) colony was established with adults collected from a greenhouse at the Thailand Science Park and released on a 1-month-old potted plant of birds eye chili (*Capsicum annuum*) cultivars TVRV 758 that were covered with a 50 × 50 × 50 cm^3^ insect screen cages. The chili thrips would eventually migrate to the new plants and build up new colonies. The thrips would reach the second larval stage within 2 weeks and be ready for laboratory assays. New plants were replaced for thrips feeding every 2 weeks.

### 2.5. Laboratory Assay

Thrips assay in the laboratory was conducted in a 100 mm diameter Petri dish with a single chili leaf that was moisturized with cotton saturated with distilled water. Assays were performed in triplicate. In each replicate, 10 2nd instar larvae were gently released onto chili leaf using a small paint brush and fungal spore suspension was sprayed by a glass nozzle sprayer with 1 bar pressure. The Petri dish was then covered with a lid. For negative control, 0.1% Tween 80 was used instead of fungal spore suspension. Distilled water was added daily for fresh leaves reservation. Mortalities were scored after 5 days of incubation at 25 °C ± 1.85% relative humidity and 12 h daily photoperiod.

### 2.6. Greenhouse Test

Two best-performing fungal strains were selected from laboratory assay to compare efficacy in greenhouse test by spray application. The experiment was conducted in an evaporative cooling greenhouse at the Thailand Science Park, starting from April to November 2015. The potted plants of birds eye chili (*C. annuum*) cultivars TVRV 758 were used as hosts for chili thrips (*S. dorsalis*). Ten adults of thrips were released per plant and the insect populations would naturally increase within 2 weeks and ready for the experiments. All spray applications were made with the knapsack sprayer.

#### 2.6.1. Comparison between *B. bassiana* (BCC48145) and *P. lilacinum* (TBRC10638)

Greenhouse tests were undertaken to compare the effects of r *B. bassiana* BCC48145 and *P. lilacinum* TBRC10638 application on the thrips population. Three treatments, including *B. bassiana* BCC48145 and *P. lilacinum* TBRC10638 spore suspension and spore diluent control, were compared using Randomized Complete Block Design (RCBD) with 3 replications. Ten potted plants infested with thrips were set for each replication in the greenhouse. The plants in each replication were sprayed with 15 mL of either fungal spore suspension containing 2 × 10^8^ spores/mL in 0.1% Tween 80 containing 0.04% APSA (AMWAY^®^) or spore diluent (0.1% Tween 80 containing 0.04% APSA) for the control. The commercial surfactant APSA was added to help spread and protect fungal spores from UV radiation. Spraying was performed using a sprayer with 3 bar pressure. Spore suspension was applied weekly for 3 consecutive weeks before the determination of the insect population by direct counting. Results from the first greenhouse test were confirmed in another repeat of the experiment.

#### 2.6.2. Median Lethal Concentration (LC_50_) Value Determination

For LC_50_ value determination, greenhouse tests using RCBD with five treatments and three replications were performed in two repeats of the experiment. Thrips-infested chili plants were prepared as described above. The treatments included spraying with 15 mL of *P. lilacinum* TBRC10638 spore suspensions at the concentrations of 2 × 10^5^, 2 × 10^6^, 2 × 10^7^ and 2 × 10^8^ spores/mL in 0.1%Tween 80 containing 0.04% APSA, and spore diluent (0.1%Tween 80 containing 0.04% APSA). Each treatment was applied in a single foliar spray for the estimation of control efficiency.

To evaluate the effects of treatments, pre-application sampling was made a day before each application and post-application sampling was made 7 days after spray application. Pre- and post-application samplings were carried out by direct counting of thrips on the three uppermost terminal leaves [28] from 30% of plants in each replication. Thrips populations were computed using the following formula to determine the control efficiency of the fungal strains according to Püntener (1981) [29].
Control efficiency = [1 − (Ta/Ca × Cb/Tb)] × 100 where
Ta = Infestation in the treatment plot after applicationTb = Infestation in the treatment plot before applicationCa = Infestation in the control plot after applicationCb = Infestation in the control plot before application

### 2.7. Field Trials

Two field trials were carried out to evaluate the effects of *P. lilacinum* TBRC10638 application on natural thrips infestation at the Tropical Vegetable Research Center (TVRC), Kasetsart University Kamphaeng Saen Campus, Nakhon Pathom Province, Thailand. Each experiment used approximately 975 m^2^ of land. One-month-old seedlings of *C. annum* cultivars TVRC 365 (spur chili) were transplanted in 20 5 × 5 m^2^ plots arranged in five columns and four rows with 2-m intervals. Each plot consisted of three 1.5 × 5 m^2^ rows, in which the seedlings were transplanted at 0.5 m apart from each other. Based on the LC_99_ values obtained from the greenhouse tests, fungal spores were prepared as suspensions containing 1.41 × 10^8^, 1.41 × 10^9^, 1.41 × 10^10^ spores/mL in 0.1% Tween 80 and 0.04% APSA. Five treatments including fungal spore suspensions at three concentrations, Imidacloprid (U dara 10^®^) diluted at the rate of 2 mL/L of water containing 0.04% APSA, and control (0.1% Tween 80 containing 0.04% APSA) were arranged in RCBD with 4 replications. The plants were sprayed one week after transplanting, and spraying was repeated weekly for 3 consecutive weeks. Spraying resumed when chili plants began to flower, once a week for 3 consecutive weeks. The same regimen was repeated at one-week intervals until harvesting. Two trials were conducted: the first trial in July-September 2017 and the second trial in October–December 2017.

Thrips populations were randomly counted from chili tips of 10 plants from each plot, 1 day before spraying and 7 days after spraying. Counting was made on a weekly basis until harvesting. Thrips populations were computed to determine the control efficiency of the fungal strains according to Püntener (1981) [28].

An assessment of the insecticidal effects of *P. lilacinum* TBRC 10638 and imidacloprid on the relative abundance of target and non-target insects present in the crop was carried out during the trials. The density of thrips (no identification), aphids, spiders, and parasitic wasps were monitored using a yellow sticky trap (7 × 11 inches). The traps were placed at both ends of plots [29]. At the end of each week for the entire cultivation period, the traps were collected for insect count and replaced by new traps. The numbers of non-target insects were counted under a stereo microscope.

### 2.8. Statistical Analysis

For laboratory assay, the corrected mortality [28] of each fungal strain was transformed by Arcsin transformation prior to one-way analysis of variance (ANOVA) (*p* value < 0.05), and means of corrected mortalities were compared using Duncan′s multiple range test ((DMRT) with SPSS11.5 program (SPSS for Windows v.11.5, IBM, www.ibm.com accessed on 1 February 2022). The strains exhibiting the highest efficacy against thrips were selected for greenhouse experiments to determine control efficiency, median lethal concentration (LC_50_), and 99% lethal concentration (LC_99_) values using Probit analysis.

For greenhouse experiments and field trials, the control efficiencies from each experiment were subjected to one-way ANOVA) (*p* value < 0.05) and mean comparison using Duncan′s multiple range test (DMRT) with an SPSS11.5 program.

## 3. Results

### 3.1. Evaluation of Entomopathogenic Fungi Insectidal Activity

Data from laboratory assay showed that *B. bassiana* BCC2660, BCC48145, and *P. lilacinum* TBRC10638 were the three most effective strains that caused 78.55%, 73.98%, and 73.93% corrected mortality against chili thrips, while *B. bassiana* BCC25950, BCC14481, BCC1658, and *I. fumosorosea* BCC1659 were second ranking with 67.01%, 65.97%, 64.06%, and 63.45% corrected mortality, respectively (*p* < 0.05, Duncan in the repeated measures model). Based on these results, *B. bassiana* BCC48145 and *P. lilacinum* TBRC10638 were selected for further cytotoxicity and greenhouse tests.

### 3.2. Cytotoxicity Results

The yield of extract from *P. lilacinum* TBRC10638 spores produced by cultivation on rice grains was 169 mg/g spore, which was less than the extract yield of 519 mg/g from *B. bassiana* BCC48145 spores. From culture filtrates, *B. bassiana* BCC48145 provided extract yields of 12.3 mg/L in static condition and 13.2 mg/L in shaken condition, which were higher than extract yields of 6.3 mg/L from static culture and 3.5 mg/L shaken culture of *P. lilacinum* TBRC10638 (Table 2). All fungal extracts were tested against KB, MCF7, NCI-H187, and GFP-expressing Vero cell lines. Results indicated that none of these extracts showed cytotoxicity toward these cell lines when tested at the concentration of 50 µg/mL (Table 2). On the other hand, the anti-cancer drugs employed as positive controls exhibited different levels of cytotoxicity towards the four cell lines (footnotes of Table 2).

### 3.3. Comparing the Efficacy of B. bassiana (BCC48145) and P. lilacinum (TBRC10638) under Greenhouse Conditions

Results from the laboratory assay showed that the fungi *B. bassiana* BCC48145 and *P. lilacinum* TBRC10638 were highly virulent against chili thrips (Figure 1). Hence, they were chosen for greenhouse applications. At the beginning of the first experiment performed in April 2015, the mean of thrips populations from the control plants at approximately 2.9 insects/Leaf was higher than those of other treatments and increased to 6 insects/Leaf in the second week of the experiment. Afterward, chili plants were destroyed by thrips before they migrated to the new plants (Figure 2A). On the plants treated with *B. bassiana* BCC48145, the thrips population started at approximately 1.2 insects/Leaf and increased to 3 insects/Leaf after one week. Nevertheless, *B**. bassiana* BCC48145 could control the thrips population in the following weeks (Figure 2A). On plants treated with *P. lilacinum* TBRC10638, the thrips population started from 2.6 insects/Leaf and gradually increased until reaching its peak at week 2 and then declined to 0.7 insects/Leaf by week 4. The control efficiency of each fungus was determined from the percentage of thrips population that was reduced after fungal treatment, using thrips population data normalized by those of the control group. Results showed that the control efficiency of *P. lilacinum* TBRC10638 (64.67%) was no different from that of *B. bassiana* BCC48145 (56.08%) (Table 3) under an average temperature of 28.33 °C ± 1.37 and humidity of 90.42% ± 7.68 in the greenhouse.

The second experiment was done in June 2015, the thrips population in all treatment groups was between 0.1–0.4 insects/plant at the start of this experiment. In the control treatment, the thrips population gradually increased until reaching 2.7 insects/Leaf in the last week (Figure 2B). On the other hand, *P. lilacinum* TBRC10638 application kept the thrips population below 0.7 insects/Leaf throughout the experiment, while *B. bassiana* BCC48145 application resulted in a thrips population of 1.3 insects/Leaf in the last week after the spraying program (Figure 2B). Consistent with results from the first experiment, the fungus *B. bassiana* BCC48145 exhibited lower control efficiency than *P. lilacinum* TBRC10638 (Table 3) under the average temperature of 28.35 °C ± 1.29 and humidity of 90.48% ± 7.57 in the greenhouse. Based on these results, the fungus *P. lilacinum* TBRC10638 was selected for the determination of LC_50_ values in the next step.

### 3.4. Determination of Lethal Concentration Values

In this experiment, the fungus *P. lilacinum* TBRC10638 at the concentration of 2 × 10^5^, 2 × 10^6^, 2 × 10^7^, 2 × 10^8^ spores/mL, and 0.1% Tween 80 were applied in a single foliar spray in order to determine the LC_50_ value. In the first experiment, starting thrips populations in different treatment groups varied from 2.4 to 8.8 insects/Leaf (Figure 3A). After application, only in the treatment using 2 × 10^6^ and 2 × 10^8^ spores/mL did the fungus exhibit an ability to control the thrips population. Chili plants from this treatment also displayed less damage from thrips than those in other groups. In contrast, thrips populations in other treatments increased in one week. The control efficiency of each treatment was calculated from thrips population data and then submitted to probit analysis. The LC_50_ value obtained from this analysis was 1.42 × 10^8^ spores/mL (Table 4) under the average temperature of 28.37 °C ± 1.45 and humidity of 83.48% ± 10.82 in the greenhouse.

In the second experiment, starting chili thrips populations in different treatment groups varied from 1.78 to 2.67 insects/Leaf (Figure 3B). All treatments showed effectiveness against thrips at 7 days after spraying, with the control efficiency of 48.70%, 56.14%, 38.70%, and 68.56% when fungal spores at the concentration of 2 × 10^5^, 2 × 10^6^, 2 × 10^7^, and 2 × 10^8^ spores/mL were applied, respectively (data not shown). The LC_50_ value obtained after probit analysis were 1.24 × 10^7^ spores/mL under the average temperature of 27.49 °C ± 1.49 and humidity of 83.85% ± 9.41 in the greenhouse. Notably, the LC_50_ value obtained from the second experiment was 10 times lower than that of the first experiment. As often the case for biological controls, the entomopathogenic fungus controls the thrips population more effectively at a lower than higher thrips infestation rate.

### 3.5. Evaluation of Fungal Spray Application in Field Trials

Thailand has a tropical climate with 3 distinct seasons; a hot season runs from March to mid-May, a rainy season from mid-May to October and a dry and relatively cool season normally runs from November to February (https://www.tmd.go.th/en/ accessed on 1 February 2022). The first field trial was conducted in the mid-rainy season (July–September 2017) with an annual rainfall of 3.4 mm (30 times/crop). The ambient temperatures were between 24.8–34.2 °C. The second field trial was conducted in the late rainy season (October–December 2017) with an average annual rainfall of 4.2 mm (22 times/crop). The ambient temperatures are between 22.8 °C and 31.0 °C.

Thrips population samples taken before spray application indicated that during the first and second field trials thrips infestation rates did not exceed 1 thrips/chili tip. After spray applications, thrips densities were maintained in both the plots treated with chemical insecticide (imidacloprid) and *P. lilacinum* as compared with the untreated control during the two trials. Till the fifth week, the fungal control efficacy ranged between 0–51.47% in the first trial and up to 90% in the second trial. There was no significant difference in the efficacy between the fungal treatments and imidacloprid treatment in both the first and second trials (F = 9.28, df = 3, *p* = 0.05). Starting from week 6 (chili flowering stage), thrips populations began to increase in all treatment plots, from 1.33–2.17 thrips/chili tip. However, there were no significant difference in the efficacy between the fungal and insecticidal treatments until the week 10 (1st trial; F = 0.170, df = 3, *p* = 0.916, 2nd trial; F = 0.113, df = 3, *p* = 0.952). An exception was found in 1 treatment that was sprayed with fungal spores at the concentration of 1.41 × 10^8^ spores/mL in the second trial, where the fungal spores were less effective than insecticide (F = 7.32, df = 4, *p* = 0.05). The control efficacy from weeks 6–10 was between 0–39.97% in the first trial and 0–54.11% in the second trial (Figure 4 and Table 5).

Application of three concentrations of *P. lilacinum* BCC10638 and imidacloprid significantly reduced the severity of damage by *S**. dorsalis* on chili as recorded by visual scores during the two trials. Different treatments showed no significant increase in yield of chili fruit during the first trial; 121.36 ± 15.98, 143.08 ± 32.32, 128.45 ± 28.49, 105.14 ± 25.55 and 137.20 ± 41.61 kg/rai (F = 0.243; df = 4, 15; *p* = 0.909) and the second trial; 293.93 ± 46.94, 277.55 ± 39.16, 303.90 ± 50.49, 344.78 ± 41.67 and 335.10 ± 42.09 kg/rai (F = 0.406; df = 4, 15; *p* = 0.801), respectively (Table 5).

Effects of fungal spores and imidaclopid application on non-target insects were monitored by using sticky traps. Results showed the presence of insect pests; thrips and aphids, natural enemies; and spiders and parasitic wasps. An increase in the number of thrips and aphids in the test area was observed after application. Moreover, there were no significant differences with 95% confidence in the densities of spiders and parasitic wasps among the different treatments. Spray application of all fungal spore suspensions and chemicals did not affect the populations of these natural enemies (Table 6). Results of both trials suggested that the natural enemies would increase their populations according to the number of pests that entered the test area.

## 4. Discussion

Screening results showed that *B. bassiana* BCC48145 and BCC2660 and *P**. lilacinum* TBRC10638 were the most virulent strains against chili thrips (*S**. dorsalis*). This was not surprising since *B**. bassiana* is generally used for insect pest control especially for insects such as chili thrips (*S**. dorsalis*) [30] and aphid (*M**. persicae*) [31], although it should be noted that the insect host of *B**. bassiana* BCC2660 was an adult of Coleopteran insect. The fungus *P**. lilacinum* TBRC10638 has been used for biocontrol of nematode pests including *Radopholus similis*, *Heterodera* spp., *Globodeera* spp. [32,33,34] and shown to kill many insect species such as *T**. palmi* in orchid farms [24], Mediterranean fruit fly (*Ceratitis capitate*), Nettle caterpillar (*Setora nitens*), Cotton aphids (*A. gossypi*), and Winchuka (*Triatoma infestans*) [35]. Thus, both *B. bassiana* and *P. lilacinum* are considered broad-spectrum fungi.

This study highlighted the potential use of entomopathogenic fungi against chili thrips (*S. dorsalis*) in the greenhouse by comparing the two fungal strains, *P. lilacinum* TBRC10638 and *B. bassiana* BCC48145, which were the two out of the three most effective strains against chili thrips according to the laboratory screening results. Although *B. bassiana* had been reported to provide effective control against thrips in the greenhouse [36], its efficacy was lower than *P. lilacinum* TBRC10638. The fungus *Purpureocillium lilacinum* is the new genus name of *Paecilomyces lilacinus* that has increasingly been reported as the causal agent of infections in man and other vertebrates [37]. Therefore, a preliminary test on the toxicity of this fungal strain toward mammalian cell lines was conducted. Results confirmed that extracts from spores and mycelial culture of *P. lilacinum* TBRC10638 displayed no cytotoxic effect on all cell lines tested, consistent with the report on *Paecilomyces lilacinus* strain 251, which is not toxic or pathogenic to mammals and approved for residential use [38]. Results from cytotoxicity tests suggested that *P. lilacinum* TBRC10638 and *B. bassiana* BCC48145 are safe to be used as thrips control.

The fungus *P. lilacinum* TBRC10638 provided better control against thrips in both greenhouse experiments than *B. bassiana* BCC48145, which showed the control efficiency of 64.67% and 72.46% for in the first and second experiment, respectively. In the 2nd experiment, due to the surge in the thrips population, the control efficacy of *B. bassiana* BCC48145 was quite low. We observed the high variability of results within and between the first and second experiments, which could be attributed to the movement of thrips between plants as reinfestation by the new migrant thrips has been described elsewhere [36].

Effective mycoinsecticides are expected to stay for a long period on the leaves and provide rapid control of the insect pest. *P. lilacinum* TBRC10638 exhibited the ability to promptly control the thrips population. According to Arthurs et al., (2013) [36], two mycoinsecticides and other bio-rational insecticides applied at 7 to 14 days intervals reduced overall *S. dorsalis* populations on pepper plants *C. annuum* cv. California Wonder in the greenhouse. *B. bassiana* GHA reduced *S. dorsalis* population by 81–94% and *I. fumosorosea* PFR-97 by 62–66%. Relatively more species and strains of entomopathogenic fungi had been tested against *F. occidentalis*. Sengonca et al., (2006) [39] tested 41 entomopathogenic fungi isolates from Thailand belonging to 25 species and 11 genera against first instar *F. occidentalis* larvae on bean leaves. Among the 14 most virulent isolates, LC_50_ values of *Beauveria* spp. ranged from 2.4 × 10^4^ to 5.9 × 10^6^ conidia/mL, *Metarhizium* spp. from 2.0 × 10^4^ to 5.0 × 10^5^ conidia/mL, and *Isaria* spp. from 3.9 × 10^4^ to 5.5 × 10^6^ conidia/mL.

In each field trial, the control efficiencies of fungi and chemicals fluctuated as the tests were conducted during the rainy season. The regular rainfall may wash away the fungal suspension, despite the addition of a commercial sticker. Control efficacies of fungal suspensions and chemicals of approximately 10–15% were not statistically different. However, more thrips mortality was observed in the second crop, which took place in the year with less rainfall. The influence of climate variation on trial results was also mentioned by Kirk (1997) and Maniania et al., (2003) [22,40].

In conclusion, mycoinsecticides may be used to control *S. dorsalis* and provide proper solutions for chemical insecticide resistance. Mycoinsecticides may be most effective in pest management programs integrated with beneficial insect pests, or in greenhouse crops where favorable environmental conditions (high humidity and low UV exposure) can be manipulated [37]. Additional research to optimize the utilization of entomopathogenic fungi—for example, through spore yield optimization, spray formulation, and storage formulation—is considered important for the improvement of insect pest control strategy.

## Figures and Tables

**Figure 1 insects-13-00684-f001:**
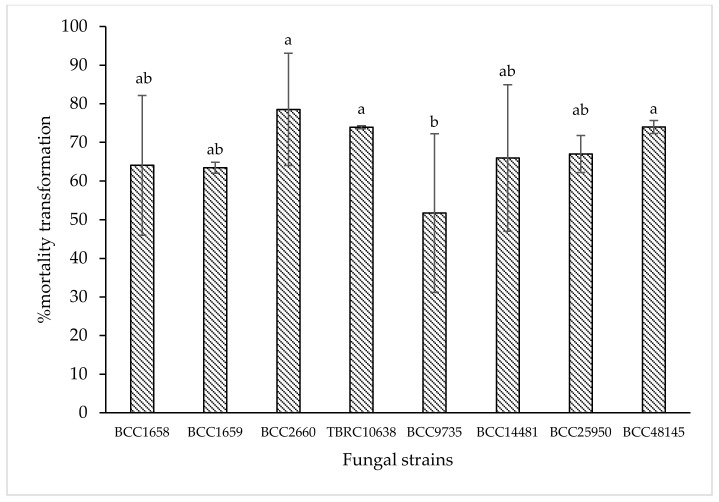
Means of % corrected mortality obtained from insect mortality assay at 5 days after treatment with 8 entomopathogenic fungi against chili thrips (*S. dorsalis*) in a laboratory. Means indicated by the same letter are not significantly different according to Duncan′s multiple range test (*p* < 0.05).

**Figure 2 insects-13-00684-f002:**
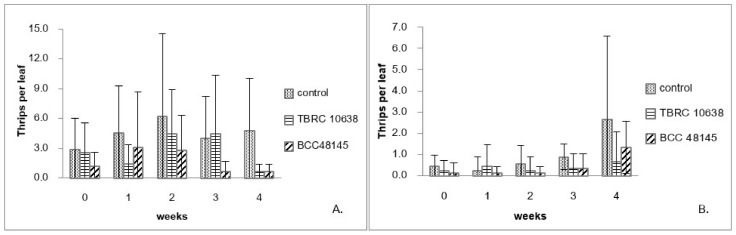
Number of chili thrips (*S. dorsalis*) on chili plants sprayed with *P. lilacinum* TBRC10638 and *B. bassiana* BCC48145 in the greenhouse. (**A**). experiment 1 (April–May 2015) and (**B**). experiment 2 (June–July 2015).

**Figure 3 insects-13-00684-f003:**
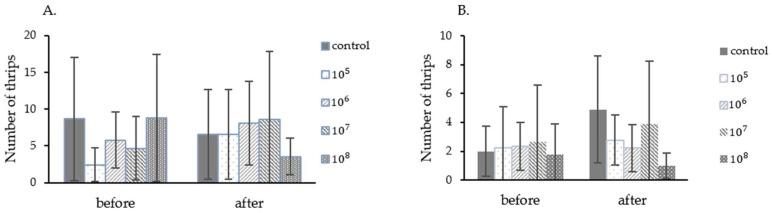
Chili thrips populations on chili plants during the determination of *P. lilacinum* TBRC10638 LC_50_ values performed in the greenhouse. (**A**). Experiment 1 (September–October 2015), (**B**). Experiment 2 (October–November 2015).

**Figure 4 insects-13-00684-f004:**
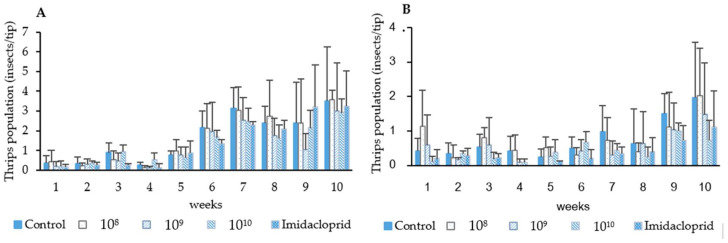
Effect of *P**. lilacinum* TBRC10638 and imidacloprid on chili thrips (*S. dorsalis*) populations during the first (**A**) and second trial (**B**).

**Table 1 insects-13-00684-t001:** List of fungal strains selected for laboratory screening against *S. dorsalis*.

Strain	Scientific Name	Insect Host of Origin	Location
BCC1658	*Beauveria bassiana*	Hymenoptera	(Kaeng Krachan National Park Phetchaburi)
BCC1659	*Isaria fumosoroseus*	Coleoptera	(Kaeng Krachan National Park Phetchaburi)
BCC 2660	*Beauveria bassiana*	Coleoptera	(Nam Nao National Park Phetchaburi)
TBRC 10638	*Purpureocilium lilacinum*	Insecta	Nakhon Ratchasrima
BCC 9735	*Metarhizium anisopliae*	Orthoptera	(Khao Yai National Park Nakhon Ratchasima)
BCC 14481	*Beauveria bassiana*	Coleoptera	(Khao Yai National Park Nakhon Ratchasima)
BCC 25950	*Beauveria bassiana*	Insecta	(Khao Yai National Park Nakhon Ratchasima)
BCC 48145	*Beauveria bassiana*	Single spore isolation from BCC2660	Pathum Thani

**Table 2 insects-13-00684-t002:** Cytotoxicity test of the extracts from *P. lilacinum* TBRC10638 and *B. bassiana* BCC48145 against 4 animal cell lines.

No.	Strains	Materials	Culture Conditions	Extract Yield	Inhibitory Concentration (IC_50_) (µg/mL)			
					KB ^1^	MCF7 ^2^	NCI-H187 ^3^	GFP-Vero ^4^
1	TBRC10638	Spores	Rice grains	169 mg/g spore	>50	>50	>50	>50
2	TBRC10638	Culture filtrate	PDB, static	6.3 mg/L	>50	>50	>50	>50
3	TBRC10638	Culture filtrate	PDB, shaken	3.5 mg/L	>50	>50	>50	>50
4	BCC48145	Spores	Rice grains	519 mg/g spore	>50	>50	>50	>50
5	BCC48145	Culture filtrate	PDB, static	12.3 mg/L	>50	>50	>50	>50
6	BCC48145	Culture filtrate	PDB, shaken	13.2 mg/L	>50	>50	>50	>50

Note: ^1^ Ellipticine (IC_50_ 3.06 µg/mL) was used as a positive control, ^2^ Tamoxifen (IC_50_ 7.83 µg/mL) and doxorubicin (IC_50_ 9.55 µg/mL) were used as positive controls, ^3^ Ellipticine (IC_50_ 7.83 µg/mL) and doxorubicin (IC_50_ 0.071 µg/mL) were used as positive controls, ^4^ Ellipticine (IC_50_ 2.88 µg/mL) was used as a positive control.

**Table 3 insects-13-00684-t003:** Control efficacy of the fungi *B. bassiana* BCC48145 and *P. lilacinum* TBRC10638 against chili thrips (*S. dorsalis*) in the greenhouse experiment.

Treatment		Experiment 1 (April–May 2015)		Experiment 2 (June–July 2015)	
		Before	After	Before	After
control	mean ± SD	2.9 ± 3.1	4.8 ± 5.2	0.4 ± 0.5	2.7 ± 3.9
TBRC10638	mean ± SD	2.6 ± 3.0	0.7 ± 0.7	0.3 ± 0.5	0.7 ± 1.4
	% efficacy	64.67 a ^1^		72.46 a	
BCC48145	mean ± SD	1.2 ± 1.4	0.7 ± 0.7	0.3 ± 0.5	1.3 ± 1.2
	% efficacy	56.08 a		30.70 ab	

Remark ^1^ Means followed by the same letter in a column are not significantly different according to Duncan′s multiple range test (*p* < 0.05).

**Table 4 insects-13-00684-t004:** Estimated lethal concentration values (spores/mL) of *P. lilacinum* TBRC10638 against chili thrips (*S. dorsalis*) in the greenhouse.

Experiment	Lethal Concentration (×10^7^ Spores/mL)		
	LC_50_	Lower-Upper Limits at 95%	Sig.
1st (September–October 2015)	14.27	8.62–243.85	<0.000 ^1^
2nd (October–November 2015)	1.24	0.43–17.51	<0.000

Remark ^1^ Since the significance level is less than 0.050, a heterogeneity factor is used in the calculation of confidence limits.

**Table 5 insects-13-00684-t005:** Control efficiency and chili yield (Kg/rai) of the chili plots sprayed with *P. lilacinum* TBRC10638 at the concentration of 1.4 × 10^8^–1.4 × 10^10^ spores/mL and imidacloprid during the first and second trials.

Times	Treatment							
	1st Crop				2nd Crop			
	1.41 × 10^8^ Spores/mL	1.41 × 10^9^ Spores/mL	1.41 × 1^10^ Spores/mL	Imidacloprid	1.41 × 10^8^ Spores/mL	1.41 × 10^9^ Spores/mL	1.41 × 10^10^ spores/mL	Imidacloprid
1st	40.29	0	0	0	60.11	52.91	0	0
2nd	20.21	40.8	10.02	50.07	0	0	52.94	45.11
3rd	14.56	23.77	0	0	33.45	64.88	42.66	90.00
4th	0	0	51.47	0	0	0	0	0
5th	27.94	18.57	14.45	41.75	57.22	27.44	20.38	29.48
6th	5.38	18.59	0	0	0	52.06	54.11	19.94
7th	0	17.5	22.28	0	25.39	0	22.58	0
8th	20.48	39.97	0	0	0	34.06	0	27.11
9th	0	0	15.21	33.91	0	0	42.14	0
Ave. (mean ± sd)	14.32 ± 4.26	17.69 ± 15.84	12.60 ± 16.80	13.97 ± 21.34	19.57 ± 25.25	25.71 ± 26.66	26.09 ± 22.71	23.52 ± 29.92
Chili yield (mean ± sd)	143.08 ± 32.32	128.45 ± 28.49	105.14 ± 25.55	137.20 ± 41.61	277.55 ± 39.16	303.90 ± 50.49	344.78 ± 41.67	335.10 ± 42.09

The results were from two independent experiments. Data shown are mean ± SD (Duncan′s multiple range test, *p* < 0.05).

**Table 6 insects-13-00684-t006:** Impact assessment of *P**. lilacinum* TBRC 10638 and imidacloprid applications on the number of non-target insects (mean ± SE) during the first and second trials.

Trial	Treatment	Thrips		Aphids		Spiders		Parasitic Wasps	
		Pre-Spray	Post-Spray	Pre-Spray	Post-Spray	Pre-Spray	Post-Spray	Pre-Spray	Post-Spray
1st	control	13.75 ± 15.72	30.25 ± 6.78	7.50 ± 9.38	8.63 ± 6.34	5.13 ± 4.92	4.25 ± 2.89	70.25 ± 54.87	81.75 ± 40.04
	1.41 × 10^8^ spores/mL	9.50 ± 9.35	41.50 ± 12.34	6.50 ± 13.04	7.38 ± 5.56	3.88 ± 2.75	3.75 ± 2.52	69.38 ± 69.51	91.63 ± 25.64
	1.41 × 10^9^ spores/mL	11.25 ± 13.48	35.88 ± 8.06	6.38 ± 7.32	9.38 ± 4.35	4.13 ± 4.43	5.63 ± 2.75	72.38 ± 59.86	90.13 ± 17.56
	1.41 × 10^10^ spores/mL	10.00 ± 8.79	35.00 ± 7.50	4.00 ± 3.16	10.75 ± 10.02	5.50 ± 3.37	5.38 ± 2.75	71.38 ± 48.57	83.75 ± 48.62
	Chemical	12.50 ± 7.79	34.13 ± 14.24	6.00 ± 7.83	7.63 ± 4.79	4.63 ± 5.74	4.00 ± 3.56	75.75 ± 89.85	91.00 ± 53.62
2nd	control	7.75 ± 8.23	23.88 ± 8.22	1.63 ± 2.22	3.50 ± 3.65	1.75 ± 2.08	13.13 ± 5.91	55.75 ± 12.61	67.63 ± 26.55
	1.41 × 10^8^ spores/mL	5.38 ± 4.03	20.50 ± 13.29	2.00 ± 1.63	2.13 ± 3.10	1.63 ± 3.30	9.13 ± 8.14	50.63 ± 53.18	57.13 ± 41.15
	1.41 × 10^9^ spores/mL	5.50 ± 9.56	24.75 ± 19.42	1.50 ± 1.41	2.13 ± 2.06	0.38 ± 0.50	15.38 ± 5.12	54.88 ± 23.96	70.00 ± 42.43
	1.41 × 10^10^ spores/mL	5.25 ± 4.65	40.50 ± 19.58	3.00 ± 2.16	3.13 ± 1.50	1.25 ± 2.38	16.63 ± 8.73	44.13 ± 17.19	74.13 ± 50.37
	Chemical	10.13 ± 4.65	18.63 ± 26.64	3.13 ± 3.86	2.38 ± 1.89	2.13 ± 3.10	12.38 ± 14.34	62.38 ± 59.06	63.38 ± 52.82

The results were from two independent experiments. Data shown are mean ± SD (Duncan′s multiple range test, *p* < 0.05).

## Data Availability

Information presented in this study can be obtained from Related authors. The information is not publicly available due to contractual obligations with research funding sources.

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
