# Peer review of "Control Efficacy of Entomopathogenic Fungus Purpureocillium lilacinum against Chili Thrips (Scirtothrips dorsalis) on Chili Plant"

_insects, 2022, doi:10.3390/insects13080684_

Round 1

Reviewer 1 Report

Review for Insects

Panyasiri et al. Control efficacy of entomopathogenic fungus Purpureocillium lilacinum against chili thrips….

This manuscript reports research to screen, select and evaluate entomopathogenic fungi (EPF) for control of thrips.  Preliminary data indicated that P. lilacinum would likely provide efficacious control of the thrips.  The manuscript reports a variety of laboratory, greenhouse and field evaluation data.   Overall, the manuscript is well organized with a logical progression of experiments.  Experimental procedures were appropriate for the most part and the data seem to be accurately reported.  Considerable grammatical errors need to be corrected throughout the manuscript.  A few are specifically noted below.

I am not familiar with the cytotoxicity tests and am content to believe the extracts showed no activity against the cell lines.  However, the entomologist in me could argue that these tests did not include a positive control to verify that the methodology would detect toxic agents.  For this reason, I tend to question the validity of these results.

Judging from the data presented, I do not believe these data satisfy assumptions for probit analysis and thus the ability to accurately report LC50 values for these pathogens.  Problems include an extremely flat slope and likely a high Chi Sq value indicating poor fit of the data to the model.  Additional comments below on this topic.

For the field trial, I appreciate the comparison with the chemical pesticide, though both provided relatively poor efficacy results. 

Suggested edits by line number.

26.  “An efficacy validation….”

52.  “ ….greenhouse cultivation where tiny insects…..”

55.  “they” is not a descriptive word in this sentence.  I suggest rewriting with descriptive terminology.

62.  Consider “Currently, integrated pests management (IPM) systems for greenhouses utilize chemical, …….”

77.  “….evaluated for control of chili thrips…..”

83.  “They proved…” Delete “were”.

87.  Correct the phrase “scratching each colonies”.  Also capitalize Tween.

90.  need to include the units with the spore concentration, presumed to be per mL.

151.  Please correct “would reached”.

153.  Was this assay consisting of three plates per treatment repeated?  Thirty exposed insects is a minimal number for evaluating for insect mortality. 

169.  Italicize the names in this sentence.

171.  This paragraph is difficult to understand and will need to be edited for clarity.  What is and “experimental set”?  I must assume that the potted plants were already infested with thrips.  It is unclear if Tween was added to the fungal treatments.  Then, why was the AmWay surfactant added and was it added to all the treatments and the control?  I assume sprays were applied “weekly” rather than continuously.  What is “before determined”?

190.  A brief description of the calculation for Control Efficiency Percentage would be helpful.  It is not intuitive from the table of results.

195.  “chili seedlings of chili” is redundant – please simplify.

212.  I am not sure of the relevance of sticky trap counts when evaluating small plots.  These data could indicate relative abundance and movement (dispersal) of winged forms of the two pest insects, foraging of spiders, and relative abundance of the parasitoids.  However, I do not thing these numbers can be directly attributed to specific treated plots.

220.  Minor statistical consideration: Were the percentage data transformed before analysis by ANOVA?  ArcSin transformation helps percentage data to fit assumptions of normality.  Also, Duncan’s MRT is not the preferred method for means separation and has generally been replaced with SNK or Tukey’s mean separation procedures. 

Figure 1.  Strain label does not match the text for the P. lilacinum. 

247.  “which was less”.

267.  Not sure I agree with the statement that P. liacinum was more efficacious than B bassiana in that the latter tended to have lower thrips densities than the former.  The calculated discrepancy may be a result of the lower time 0 value for the Bb fungus.

287.  The text about the Lethal Concentration values determination leaves me with many questions.  I do not know how movement of the thrips among treated plants was prevented.  I am constantly questioning the impact for the potential re-infestation of treated plants used in these experiments.  Further, The calculations used to determine mortality percentages is not clear form the text.  (reference to Puntener 1981).  Data presented in Figure 3 do not provide a clear dosage response trend.  Further, the LC50 value for the first experiment reported in table 4 is beyond the range of spore concentrations applied to the plants.  This suggests a poor fit of the data to the probit model.  I believe a more controlled exposure/evaluation techniques is needed to reduce the variability reported between these two experiment. 

335.  Reporting of ANOVA statistical data is incorrect.  ANOVA requires two values for the DF.  The proper numerator DF for the treatments should be 4 (5 treatments minus 1 =4).  The numerator DF for comparing among replications should be 3 (4reps minus 1 =3).  The denominator DF is the error DF in the table, presumably 12 based on the RCB design and treatments.  Please review and revise as needed the presentation of the ANOVA results in this text.

Table 5, what are the units for “Times”?  Weeks?

370, “highest strains”?  Do you mean most virulent?

371.  change to “not surprising”

Delete “homopteran”.

396.  Thanks for noting this inability to control insect movement.  Might be helpful to note how to address this problem for future research with thrips.

408.  These LC50 values are more in line with results I expect from laboratory experiments.  Here LC50 values were based on GH experiments with poor control of pest movement, hence higher values were reported. 

I did not specifically evaluate references cited other than to note format differences among those reported in the references section.

End

Author Response

Responses to reviewers' comments

Reviewer 1

This manuscript reports research to screen, select and evaluate entomopathogenic fungi (EPF) for control of thrips.  Preliminary data indicated that P. lilacinum would likely provide efficacious control of the thrips.  The manuscript reports a variety of laboratory, greenhouse, and field evaluation data.   Overall, the manuscript is well organized with a logical progression of experiments.  Experimental procedures were appropriate for the most part and the data seem to be accurately reported.  Considerable grammatical errors need to be corrected throughout the manuscript.  A few are specifically noted below.

  1. Was this assay consisting of three plates per treatment repeated?  Thirty exposed insects is a minimal number for evaluating for insect mortality.

Response:

For laboratory assay, we used ten 2nd instar larvae for each replication and 3 replications for each treatment.Therefore, the number of insects used for mortality evaluation was 30 insects per treatment.

  1. I am not sure of the relevance of sticky trap counts when evaluating small plots.These data could indicate relative abundance and movement (dispersal) of winged forms of the two pest insects, foraging of spiders, and relative abundance of the parasitoids.  However, I do not thing these numbers can be directly attributed to specific treated plots.

Response:

We agree with this comment, as we also saw the increase of natural enemy populations when the pests entered the test area. Additionally, we did not see any significant difference in the numbers of these insects between treatments.

  1. Minor statistical consideration: Were the percentage data transformed before analysis by ANOVA?  ArcSin transformation helps percentage data to fit assumptions of normality.Also, Duncan’s MRT is not the preferred method for means separation and has generally been replaced with SNK or Tukey’s mean separation procedures. 

Response:

As suggested by the reviewer, we have applied ArcSin transformation to percentage data and used the transformed data for means separation analyses using both Tukey's and DMRT methods. As a result, data from laboratory assay show no difference between treatments when analyzed by the Tukey's method, whereas DMRT analysis separates data into two groups. For the rest of the experiments, both Tukey's and DMRT methods revealed no difference between treatments.

  1. The text about the Lethal Concentration values determination leaves me with many questions. I do not know how the movement of the thrips among treated plants was prevented.  I am constantly questioning the impact for the potential re-infestation of treated plants used in these experiments.  Further, The calculations used to determine mortality percentages are not clear from the text.  (reference to Puntener 1981).  Data presented in Figure 3 do not provide a clear dosage response trend.  Further, the LC50 value for the first experiment reported in table 4 is beyond the range of spore concentrations applied to the plants.  This suggests a poor fit of the data to the probit model.  I believe more controlled exposure/evaluation techniques are needed to reduce the variability reported between these two experiments. 

Response:

The experiments for LC determination were performed in a cooling greenhouse divided by 2-meter high partition walls to separate between treatments. These by no means prevented insect movement among treated plants.  Also, there might be a small chance of external thrips entering through the cooling system. Therefore, we believe that thrips migration contributes to the discrepancies of data including the lack of dose response.

To address reviewer's question on the calculation of data, we provide more details on the calculation of control efficiency under section 2.6.2 of Materials and methods.

To address reviewer's comment on the LC values that exceed the range of spore concentrations, we decide not to report the LC99 values. We agree with the reviewer's comment on the variability of results and the poor fit of data, and that a more controlled method should be employed for the determination of LC values. 

  1. Thanks for noting this inability to control insect movement.Might be helpful to note how to address this problem for future research with thrips.

Response:

We agree that controlling insect movement would be a challenging task. One of the solutions that we may adopt in the future is switching to cage experiments where individual plants are placed under cages covered with fine mesh materials.

  1. These LC50 values are more in line with results I expect from laboratory experiments. Here LC50 values were based on GH experiments with poor control of pest movement, hence higher values were reported. 

Response:

We acknowledge this comment from reviewer.

Reviewer 2 Report

This is an original and interesting article. The methodology used for this review is properly and clearly described. Data have been correctly interpreted and conclusions are sound. However, the standard deviation of Figure 2 and table3 seems very high, the authors should explain it. The grammatical quality of the text also needs substantial improvement.

There are many errors in the manuscript, the author must correct these errors very carefully, for instance:

line 147: change “S. Dorsalis” to “S. dorsalis

line 169: “B. bassiana” and “P. lilacinum” must be italic,

line 236: What does “I. fumosorosea BCC1659” refer to?

line 304: which figure is “(FigureB).” stand for?

Author Response

Responses to reviewers' comments

Reviewer 2

This is an original and interesting article. The methodology used for this review is properly and clearly described. Data have been correctly interpreted and conclusions are sound. However, the standard deviation of Figure 2 and table3 seems very high, the authors should explain it. The grammatical quality of the text also needs substantial improvement.

Response:

The greenhouse tests were performed with thrips infested plants, on which the insects were released and allowed to populate for two weeks.  Therefore, it was not possible to control starting thrips populations in different treatment groups to begin with.  We believe that the migration of thrips after the damage of control plants also contributed to the high variation of results.

line 304: which figure is “(FigureB).” stand for?

Response:

The referred text has been corrected to "(Figure 3B)".